# Simulating a Primary Visual Cortex at the Front of CNNs Improves Robustness to Image Perturbations

**Joel Dapello**[*,1,2,3], **Tiago Marques**[*,1,2,4]
**Martin Schrimpf**[1,2,4], **Franziska Geiger**[2,5,6,7], **David D. Cox**[8,3], **James J. DiCarlo**[1,2,4]

[*]Joint first authors (equal contribution)
[1]Department of Brain and Cognitive Sciences, MIT, Cambridge, MA02139
[2]McGovern Institute for Brain Research, MIT, Cambridge, MA02139
[3]School of Engineering and Applied Sciences, Harvard University, Cambridge, MA02139
[4]Center for Brains, Minds and Machines, MIT, Cambridge, MA02139
[5]University of Augsburg
[6]Ludwig Maximilian University
[7]Technical University of Munich
[8]MIT-IBM Watson AI Lab

dapello@mit.edu      tmarques@mit.edu

## Abstract

Current state-of-the-art object recognition models are largely based on convolutional neural network (CNN) architectures, which are loosely inspired by the primate visual system. However, these CNNs can be fooled by imperceptibly small, explicitly crafted perturbations, and struggle to recognize objects in corrupted images that are easily recognized by humans. Here, by making comparisons with primate neural data, we first observed that CNN models with a neural hidden layer that better matches primate primary visual cortex (V1) are also more robust to adversarial attacks. Inspired by this observation, we developed VOneNets, a new class of hybrid CNN vision models. Each VOneNet contains a fixed weight neural network front-end that simulates primate V1, called the VOneBlock, followed by a neural network back-end adapted from current CNN vision models. The VOneBlock is based on a classical neuroscientific model of V1: the linear-nonlinear-Poisson model, consisting of a biologically-constrained Gabor filter bank, simple and complex cell nonlinearities, and a V1 neuronal stochasticity generator. After training, VOneNets retain high ImageNet performance, but each is substantially more robust, outperforming the base CNNs and state-of-the-art methods by 18% and 3%, respectively, on a conglomerate benchmark of perturbations comprised of white box adversarial attacks and common image corruptions. Finally, we show that all components of the VOneBlock work in synergy to improve robustness. While current CNN architectures are arguably brain-inspired, the results presented here demonstrate that more precisely mimicking just one stage of the primate visual system leads to new gains in ImageNet-level computer vision applications.

## 1 Introduction

For the past eight years, convolutional neural networks (CNNs) of various kinds have dominated object recognition [1, 2, 3, 4], even surpassing human performance in some benchmarks [5]. However, scratching beneath the surface reveals a different picture. These CNNs are easily fooled by imperceptibly small perturbations explicitly crafted to induce mistakes, usually referred to as adversarial

attacks [6, 7, 8, 9, 10]. Further, they exhibit a surprising failure to recognize objects in images corrupted with different noise patterns that humans have no trouble with [11, 12, 13]. This remarkable fragility to image perturbations has received much attention in the machine learning community, often from the perspective of safety in real-world deployment of computer vision systems [14, 15, 16, 17, 18, 19, 20, 21, 22]. As these perturbations generally have no perceptual alignment with the object class [23], the failures suggest that current CNNs obtained through task-optimization end up relying on visual features that are not all the same as those used by humans [24, 25]. Despite these limitations, some CNNs have achieved unparalleled success in partially explaining neural responses at multiple stages of the primate ventral stream, the set of cortical regions underlying primate visual object recognition [26, 27, 28, 29, 30, 31, 32].

**How can we develop CNNs that robustly generalize like human vision?** Incorporating biological constraints into CNNs to make them behave more in line with primate vision is an active field of research [33, 34, 35, 36, 37, 38, 39, 40, 41, 42, 43, 32]. Still, no neurobiological prior has been shown to considerably improve CNN robustness to both adversarial attacks and image corruptions in challenging real-world tasks such as ImageNet [44]. Here, we build on this line of work, starting with the observation that the ability of each CNN to explain neural response patterns in primate primary visual cortex (V1) is strongly correlated with its robustness to imperceptibly small adversarial attacks. That is, the more biological a CNN's "V1" is, the more adversarially robust it is.

Inspired by this, we developed *VOneNets*, a new class of hybrid CNNs, containing a biologically-constrained neural network that simulates primate V1 as the front-end, followed by an off-the-shelf CNN back-end trained using standard methods. The V1 front-end, *VOneBlock*, is based on the classical neuroscientific linear-nonlinear-Poisson (LNP) model, consisting of a fixed-weight Gabor filter bank (GFB), simple and complex cell nonlinearities, and neuronal stochasticity. The VOneBlock outperforms all standard ImageNet trained CNNs we tested at explaining V1 responses to naturalistic textures and noise samples. After training, VOneNets retain high ImageNet performance, but are substantially more robust than their corresponding base models, and compete with state-of-the-art defense methods on a conglomerate benchmark covering a variety of adversarial images and common image corruptions. Importantly, these benefits transfer across different architectures including ResNet50 [4], AlexNet [1], and CORnet-S [32]. We dissect the VOneBlock, showing that all properties work in synergy to improve robustness and that specific aspects of VOneBlock circuitry offer robustness to different perturbation types. Notably, we find that neuronal stochasticity plays a large role in the white box adversarial robustness, but that stochasticity alone is insufficient to explain our results—neuronal stochasticity interacts supralinearly with the VOneBlock features to drive adversarial robustness. Finally, as a large percentage of this robustness remains even when we remove stochasticity only during the adversarial attack, we conclude that training with stochasticity at the VOneBlock level leads the downstream layers to learn more robust representations.

Model weights and code are available at `https://github.com/dicarlolab/vonenet`.

## 1.1 Related Work

**V1 modeling** Since the functional characterization of simple and complex cell responses by Hubel and Wiesel [45], modeling V1 responses has been an area of intense research. Early approaches consisting of hand-designed Gabor filters were successful in predicting V1 simple cell [46] and complex cell [47] responses to relatively simple stimuli. These models were later improved with the addition of further nonlinear operations, such as normalization and pooling, to account for extra-classical functional properties [48, 49]. Early hierarchical models of object recognition, which incorporated these type of V1 models as their early layers, were used with some success in modeling neuronal responses in the ventral stream and object recognition behavior [50, 51]. Generalized LNP models expanded the V1 model class by allowing a set of fitted excitatory and suppressive spatial filters to be nonlinearly combined [52], and subunit models introduced two sequential linear-nonlinear (LN) stages for fitting V1 responses [53]. Recently, both task-optimized and data-fitted CNNs were shown to narrowly beat a GFB model in predicting V1 responses, further validating that multiple LN stages may be needed to capture the complexity of V1 computations [30].

**Model robustness** Much work has been devoted to increasing model robustness to adversarial attacks [54, 17, 15, 55, 56, 40, 57], and to a lesser extent common image corruptions [13, 58, 59]. In the case of adversarial perturbations, the current state-of-the-art is adversarial training, where a network is explicitly trained to correctly classify adversarially perturbed images [17]. Adversarial training

is computationally expensive [60, 61], known to impact clean performance [62], and overfits to the attack constraints it is trained on [63, 64]. Other defenses involve adding noise either during training [59], inference [65, 19], or both [15, 66, 67]. In the case of stochasticity during inference, Athalye et. al. demonstrated that fixing broken gradients or taking the expectation over randomness often dramatically reduces the effectiveness of the defense [68]. In a promising demonstration that biological constraints can increase CNN robustness, Li et. al. showed that biasing a neural network's representations towards those of the mouse V1 increases the robustness of grey-scale CIFAR [69] trained neural networks to both noise and white box adversarial attacks [40].

## 2 Adversarial Robustness Correlates with V1 Explained Variance

The susceptibility of current CNNs to be fooled by imperceptibly small adversarial perturbations suggests that these CNNs rely on some visual features not used by the primate visual system. Are models that better explain neural responses in the macaque V1 more robust to adversarial attacks? We analyzed an array of publicly available neural networks with standard ImageNet training [70] including AlexNet [1], VGG [3], ResNet [4], ResNeXt [71], DenseNet [72], SqueezeNet [73], ShuffleNet [74], and MnasNet [75], as well as several ResNet50 models with specialized training routines, such as adversarial training with $L_\infty$ ($\|\delta\|_\infty = 4/255$ and $\|\delta\|_\infty = 8/255$) and $L_2$ ($\|\delta\|_2 = 3$) constraints [76], and adversarial noise combined with Stylized ImageNet training [59].

For each model, we evaluated how well it explained the responses of single V1 neurons evoked by given images using a standard neural predictivity methodology based on partial least square regression (PLS) [31, 32]. We used a neural dataset with 102 neurons and 450 different 4deg images, consisting of naturalistic textures and noise samples [77]. Explained variance was measured using a 10-fold cross-validation strategy. Similarly to other studies, we considered the field-of-view of all models to span 8deg [31, 30] (see Supplementary Section A for more details). To evaluate the adversarial robustness of each model in the pool, we used untargeted projected gradient descent (PGD) [17], an iterative, gradient-based white box adversarial attack with $L_\infty$, $L_2$, and $L_1$ norm constraints of $\|\delta\|_\infty = 1/1020$, $\|\delta\|_2 = 0.15$, and $\|\delta\|_1 = 40$, respectively. For each norm, the attack strength was calibrated to drive variance in performance amongst non-adversarially trained models, resulting in perturbations well below the level of perceptibility (see Supplementary Section B.1 for more details).

We found that accuracy under these white box adversarial attacks has a strong positive correlation with V1 explained variance (Fig. 1, r=0.85, p=2.1E-9, n=30). Notably, adversarially trained ResNet50 models [76], which were hardly affected by these attacks, explained more variance in V1 neural activations than any non-adversarially trained neural network. The correlation was not driven by the CNNs' clean ImageNet performance since it was even more pronounced when the white box accuracy was normalized by the clean accuracy (r=0.94, p=7.4E-15, n=30) and was also present when removing the adversarially trained models (r=0.73, p=1.78E-5, n=27). While increasing the perturbation strength rapidly decreases the accuracy of non-adversarially trained models, the described correlation was present for a wide range of attack strengths: when the perturbation was multiplied by a factor of 4, greatly reducing the variance of non-adversarially trained models, white box accuracy was still significantly correlated with explained variance in V1 (r=0.82, p=1.17E-8, n=30).

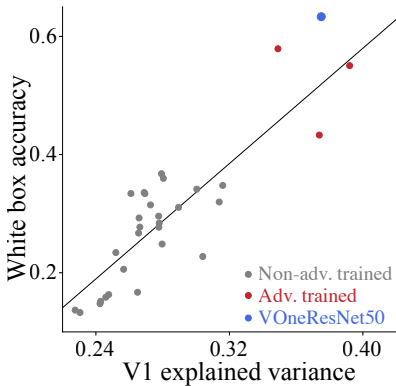

Figure 1: **CNNs' robustness to white box attacks correlates with explained response variance in primate V1.** Comparison of top-1 accuracy under white box attacks of low perturbation strengths (average of 3 PGD constraints: $\|\delta\|_\infty = 1/1020$, $\|\delta\|_2 = 0.15$, and $\|\delta\|_1 = 40$) against fraction of explained variance of V1 responses (using PLS regression) for a pool of CNN models. Perturbation strength was chosen to drive variance across model performance. White box accuracy and V1 explained variance are significantly correlated (r=0.85, p=2.1E-9, n=30 CNNs, linear fit shown in gray line). Gray circles, non-adversarially trained trained CNNs (n=27); red circles, adversarially trained ResNet50 models (n=3); blue circle, VOneResNet50 (not included in correlation).

# 3   VOneNet: a Hybrid CNN with a V1 Neural Network Front-End

Inspired by the strong correlation between V1 explained variance and robustness to white box attacks, we developed the VOneNet architecture. The major characteristic that sets the VOneNet architecture apart is its V1 front-end, the VOneBlock. While most of its layers have parameters learned during ImageNet training, VOneNet's first block is a fixed-weight, mathematically parameterized CNN model that approximates primate neural processing of images up to and including area V1. Importantly, the VOneBlock components are mapped to specific neuronal populations in V1, which can be independently manipulated or switched off completely to evaluate their functional role. Finally, the VOneBlock can be easily adapted to different CNN base architectures as described below. Here we build VOneNets from three base architectures: ResNet50 [4], CORnet-S [32], and AlexNet [1].

A VOneNet consists of the VOneBlock and a back-end network adapted from a base CNN (Fig. 2). When building a VOneNet from a particular CNN, we replace its first block (typically one stack of convolutional, normalization, nonlinear, and pooling layers, Supplementary Table C.3) by the VOneBlock and a trained transition layer. The VOneBlock matches the replaced block's spatial map dimensions ($56 \times 56$ for the base CNNs considered) but can have more channels (for the standard model we chose $C_{V1} = 512$; see Supplementary Fig. C.2 for an analysis of how the number of channels affects the results). It is followed by the transition layer, a $1 \times 1$ convolution, that acts as a bottleneck to compress the higher channel number to the original block's depth. The VOneBlock is inspired by the LNP model of V1 [52], consisting of three consecutive processing stages—convolution, nonlinearity, and stochasticity generator—with two distinct neuronal types— simple and complex cells—each with a certain number of units per spatial location (standard model with $SC_{V1} = CC_{V1} = 256$). The following paragraphs describe the main components of the VOneBlock (see Supplementary Section C for a more detailed description).

**Biologically-constrained Gabor filter bank** The first layer of the VOneBlock is a mathematically parameterized GFB with parameters tuned to approximate empirical primate V1 neural response data. It convolves the RGB input images with Gabor filters of multiple orientations, sizes, shapes, and spatial frequencies, in a $56 \times 56$ spatial map. To instantiate a VOneBlock, we randomly sample $C_{V1}$ values for the GFB parameters according to empirically observed distributions of preferred orientation, peak spatial frequency, and size/shape of receptive fields [78, 79, 80]. The VOneBlock keeps color processing separate, with each Gabor filter convolving a single color channel from the input image. The resulting set of spatial filters is considerably more heterogeneous than those found in the first layer of standard CNNs, and better approximates the diversity of primate V1 receptive fields (Supplementary Fig. C.1).

**Simple and complex cells** While simple cells were once thought to be an intermediate step for computing complex cell responses, it is now known that they form the majority of downstream projections to V2 [81]. For this reason, the VOneBlock nonlinear layer has two different nonlinearities that are applied to each channel depending on its cell type: a rectified linear transformation for simple cells, and the spectral power of a quadrature phase-pair for complex cells.

**V1 stochasticity** A defining property of neuronal responses is their stochasticity—repeated measurements of a neuron in response to nominally identical visual inputs results in different spike trains. In awake monkeys, the mean spike count (averaged over many repetitions) depends on the presented image, and the spike train for each trial is approximately Poisson: the spike count variance is equal to the mean [82]. To approximate this property of neuronal responses, we add independent Gaussian noise to each unit of the VOneBlock, with variance equal to its activation. Before doing this, we apply an affine transformation to the units' activations so that both the mean stimulus response and the mean baseline activity are the same as those of a population of primate V1 neurons measured in a 25ms time-window (mean stimulus response and spontaneous activity of 0.324 and 0.073 spikes, respectively; see Supplementary Table C.2 and Fig. C.2 for other time-windows). Like in the brain, the stochasticity of the VOneBlock is always on, during both training and inference.

**V1 explained variance** The VOneBlock was not developed to compete with state-of-the-art data-fitting models of V1 [53, 30]. Instead we used available empirical distributions to constrain a GFB model, generating a neuronal space that approximates that of primate V1. Despite its simplicity, the VOneBlock outperformed all tested standard ImageNet trained CNNs in explaining responses in the V1 dataset used, and with an explained variance of 0.375±0.006, came right in the middle of the range of the adversarially trained CNNs (Fig. 1, blue circle). On the surface, it seems that these results are

at odds with Cadena et al which showed that a task-optimized CNN marginally outperformed a GFB model in explaining V1 responses [30]. However, our GFB has parameters constrained by empirical data resulting in a better model of V1; when we use the parameters of the GFB in Cadena et al, we obtained a much lower explained variance (0.296±0.005). Our results suggest that a well-tuned GFB and adversarially trained CNNs are currently the best models in predicting primate V1 responses.

## 4 Results

### 4.1 Simulating a V1 at the front of CNNs improves robustness to white box attacks

To evaluate the VOneNets' robustness to whitebox adversarial attacks, we used untargeted PGD with $L_\infty$, $L_2$, and $L_1$ norm constraints of $\|\delta\|_\infty \in [1/1020, 1/255]$, $\|\delta\|_2 \in [0.15, 0.6]$, and $\|\delta\|_1 \in [40, 160]$. All VOneNets were attacked end-to-end, with gradients propagated through the VOneBlock to the pixels. Because the VOneBlock is stochastic, we took extra care when attacking VOneNets. We use the reparameterization trick when sampling VOneBlock stochasticity, allowing us to keep the gradients intact [83]. Further, to combat the effects of noise in the gradients, we adapted our attack such that at every PGD iteration, we take 10 gradient samples and move in the average direction to increase the loss [68] (Supplementary Table B.1). To verify the effectiveness of our attacks, we used controls suggested by Carlini et. al. 2019 [84]; in particular, we show that increasing the magnitude of the norm of the attack monotonically decreases VOneNet accuracy, eventually reaching 0% accuracy (Supplementary Fig. B.1). Additionally, increasing the number of PGD steps from 1 to

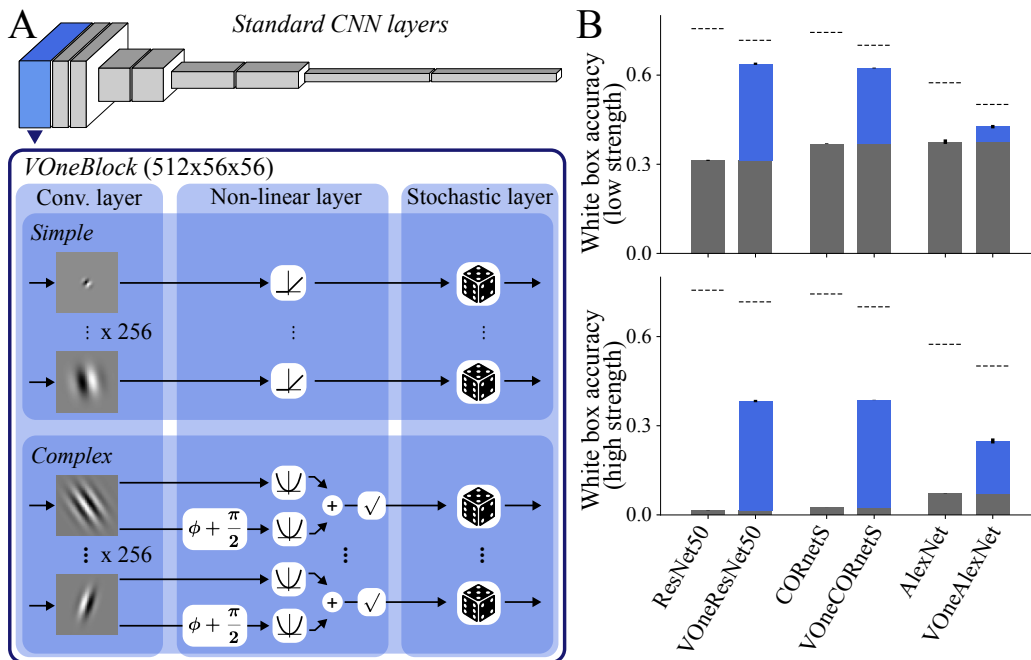

Figure 2: **Simulating V1 at the front of CNNs improves robustness to white box attacks. A** A VOneNet consists of a model of primate V1, the VOneBlock, followed by a standard CNN architecture. The VOneBlock contains a convolutional layer (a GFB with fixed weights constrained by empirical data), a nonlinear layer (simple or complex cell nonlinearities), and a stochastic layer (V1 stochasticity generator with variance equal to mean). **B** Top, comparison of accuracy (top-1) on low strength white box attacks (PGD with $\|\delta\|_\infty = 1/1020$, $\|\delta\|_2 = 0.15$, $\|\delta\|_1 = 40$) between three base models (ResNet50, CORnet-S, and AlexNet) and their correspoding VOneNets. Gray bars show the performance of base models. Blue bars show the improvements from the VOneBlock. Dashed lines indicate the performance on clean images. Bottom, same but for white box attacks of higher strength (PGD with $\|\delta\|_\infty = 1/255$, $\|\delta\|_2 = 0.6$, $\|\delta\|_1 = 160$). VOneNets show consistently higher white box accuracy even for perturbation strengths that reduce the performance of the base models to nearly chance. Error-bars represent SD (n=3 seeds).

many strictly increases the effect of the attack under a given norm size (Supplementary Fig. B.2). Both of these indicate that VOneNet gradients contain useful information for the construction of adversarial examples. For more details and controls, see Supplementary Sections B.1 and B.2.

We found that simulating a V1 front-end substantially increased the robustness to white box attacks for all three base architectures that we tested (ResNet50, CORnet-S, and AlexNet) while leaving clean ImageNet performance largely intact (Fig. 2 B). This was particularly evident for the stronger perturbation attacks, which reduced the accuracy of the base models to nearly chance (Figs. 2 B bottom). This suggests that the VOneBlock works as a generic front-end, which can be transferred to a variety of different neural networks as an architectural defense to increase robustness to adversarial attacks.

## 4.2 VOneNets outperform state-of-the-art methods on a composite set of perturbations

We then focused on the ResNet50 architecture and compared VOneResNet50 with two state-of-the-art training-based defense methods: adversarial training with a $\|\delta\|_\infty = 4/255$ constraint ($AT_{L_\infty}$) [76], and adversarial noise with Stylized ImageNet training ($ANT^{3\times3}$+SIN) [59]. Because white box adversarial attacks are only part of the issue of model robustness, we considered a larger panel of image perturbations containing a variety of common corruptions. For evaluating model performance on corruptions we used the ImageNet-C dataset [13] which consists of 15 different corruption types, each at 5 levels of severity, divided into 4 categories: noise, blur, weather, and digital (see Supplementary Fig. B.5 for example images and Supplementary Section B.3 for more details).

As expected, each of the defense methods had the highest accuracy under the perturbation type that it was designed for, but did not considerably improve over the base model on the other (Table 1). While the $AT_{L_\infty}$ model suffered substantially on corruptions and clean performance, the $ANT^{3\times3}$+SIN model, the current state-of-the-art for common corruptions, had virtually no benefit on white box attacks. On the other hand, VOneResNet50 improved on both perturbation types, outperforming all the models on perturbation mean (average of white box and common corruptions) and overall mean (average of clean, white box, and common corruptions), with a difference to the second best model of 3.3% and 5.3%, respectively. Specifically, VOneResNet50 showed substantial improvements for all the white box attack constraints (Supplementary Fig. B.1), and more moderate improvements for common image corruptions of the categories noise, blur, and digital (Supplementary Table B.2 and Fig. B.6). These results are particularly remarkable since VOneResNet50 was not optimized for robustness and does not benefit from any computationally expensive training procedure like the other defense methods. When compared to the base ResNet50, which has an identical training procedure, VOneResNet50 improves 18% on perturbation mean and 10.7% on overall mean (Table 1).

Table 1: **VOneResNet50 outperforms other defenses on perturbation mean and overall mean**. Accuracy (top-1) on overall mean, perturbation mean, white box attacks, common corruptions, and clean images for standard ResNet50 and three defense methods: adversarial training, adversarial noise combined with Stylized ImageNet training, and VOneResNet50. Perturbation mean is the average accuracy over white box attacks and common image corruptions. The overall mean is the average accuracy over the two perturbation types and clean ImageNet. Values are reported as mean and SD (n=3 seeds).

| | Overall | Perturbation | | | |
| | Mean [%] | Mean [%] | White box [%] | Corruption [%] | Clean [%] |
|---|---|---|---|---|---|
| Base ResNet50 [4] | $43.6_{\pm0.0}$ | $27.6_{\pm0.1}$ | $16.4_{\pm0.1}$ | $38.8_{\pm0.3}$ | $\mathbf{75.6}_{\pm0.1}$ |
| $AT_{L_\infty}$ [76] | 49.0 | 42.3 | **52.3** | 32.3 | 62.4 |
| $ANT^{3\times3}$+SIN [59] | 48.0 | 34.9 | 17.3 | **52.6** | 74.1 |
| **VOneResNet50** | $\mathbf{54.3}_{\pm0.1}$ | $\mathbf{45.6}_{\pm0.2}$ | $51.1_{\pm0.4}$ | $40.0_{\pm0.3}$ | $71.7_{\pm0.1}$ |

## 4.3 All components of the VOneBlock contribute to improved model robustness

Since the VOneBlock was not explicitly designed to increase model robustness, but rather to approximate primate V1, we investigated which of its components are responsible for the increased

robustness we observe. We performed a series of experiments wherein six new VOneNet variants were created by removing or modifying a part of the VOneBlock (referred to as VOneBlock variants). After ImageNet training, model robustness was evaluated as before on our panel of white box attacks and image corruptions. Three variants targeted the GFB: one sampling the Gabor parameters from uniform distributions instead of those found in primate V1, another without high spatial frequency (SF) filters ($f < 2$cpd), and another without low SF filters ($f > 2$cpd). Two additional variants targeted the nonlinearities: one without simple cells, and another without complex cells. Finally, the sixth variant had the V1 neuronal stochasticity generator removed. Even though all VOneBlock variants are poorer approximations of primate V1, all resulting VOneResNet50 variants still had improved perturbation accuracy when compared to the base ResNet50 model (Fig. 3 A). On the other hand, all variants except that with uniformly sampled Gabor parameters showed significant deficits in robustness compared to the unmodified VOneResNet50 (Fig. 3 A, drops in perturbation accuracy between 1% and 15%).

Interestingly, we observed that some of these changes had a highly specific effect on the type of perturbation robustness affected (Table 2 and Supplementary Section D). Removing high SF Gabors negatively affected both white box and clean accuracy while actually improving robustness to common image corruptions, particularly those of the noise and blur categories (Supplementary Table D.1). Removing complex cells only impaired white box robustness, as opposed to removing simple cells, which was particularly detrimental to performance on image corruptions. Finally, removing V1 stochasticity considerably decreased white box accuracy while improving accuracy for both clean and corrupted images.

The VOneBlock variant without V1 stochasticity suffered the most dramatic loss in robustness. This is not altogether surprising, as several approaches to adversarial robustness have focused on noise as a defense [15, 19, 66]. To investigate whether V1 stochasticity alone accounted for the majority of the robustness gains, we evaluated the perturbation accuracy of a ResNet50 model with V1 stochasticity added at the output of its first block. Neuronal stochasticity was implemented exactly the same way as in the VOneBlock by first applying an affine transformation to scale the activations so that they match primate V1 neuronal activity. Like the VOneResNet50, this model had stochasticity during training and inference, and showed a considerable improvement in robustness compared

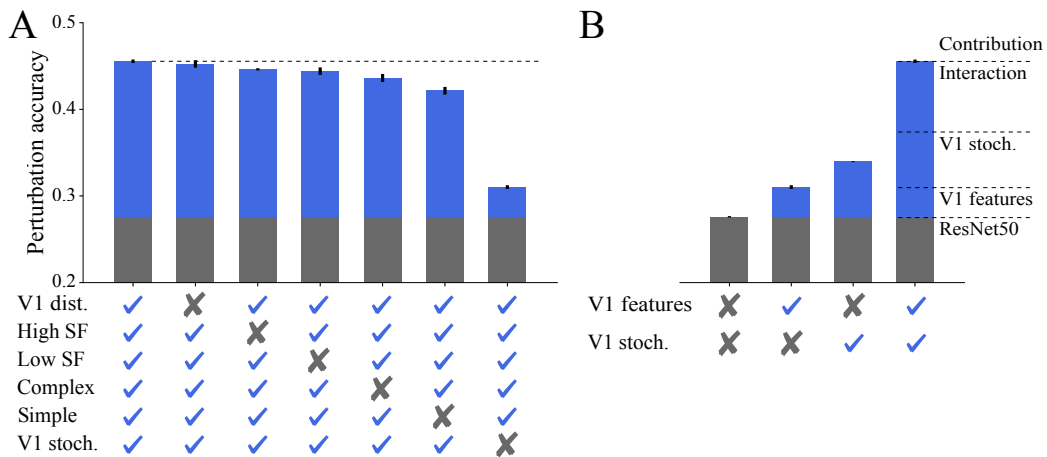

Figure 3: **All components of the VOneBlock work in synergy to improve robustness to image perturbations. A** Perturbation mean accuracy (top-1) for VOneResNet50 and several variations with a component of the VOneBlock removed or altered. From left to right: unmodified VOneResNet50, model with Gabor parameters sampled uniformly (within biological ranges), model without high SF Gabors, model without low SF Gabors, model without complex cells, model without simple cells, model without V1 stochasticity. Gray bars show the performance of ResNet50. Blue bars show the improvements due to the VOneBlock. Dashed line indicates the accuracy of the unmodified model. Error-bars represent SD (n=3 seeds). **B** Same as in **A** but comparing ResNet50, VOneResNet50 without V1 stochasticity, ResNet50 with V1 stochasticity added after the first block, and VOneResNet50. Adding V1 stochasticity to ResNet50 accounts for less than half of the total improvement of VOneResNet50, demonstrating a supralinear interaction between the V1 features and V1 stochasticity.

Table 2: **Removal of some VOneBlock components causes impairments with high specificity.** Difference in accuracy (top-1) relative to the unmodified VOneResNet50, for each of the variants with removed components on overall mean, white box attacks, common corruptions, and clean images. Values are reported as mean and SD (n=3 seeds).

| Component removed | Overall Mean [$\Delta\%$] | Perturbation White box [$\Delta\%$] | Corruption [$\Delta\%$] | Clean [$\Delta\%$] |
|---|---|---|---|---|
| V1 dist. | $-0.4_{\pm 0.3}$ | $0.4_{\pm 0.8}$ | $-1.1_{\pm 0.3}$ | $-0.5_{\pm 0.3}$ |
| High SF | $-2.2_{\pm 0.2}$ | $-3.8_{\pm 0.3}$ | $1.9_{\pm 0.4}$ | $\mathbf{-4.7}_{\pm 0.2}$ |
| Low SF | $-0.7_{\pm 0.5}$ | $-1.6_{\pm 1.1}$ | $-0.7_{\pm 0.1}$ | $0.1_{\pm 0.3}$ |
| Complex | $-1.0_{\pm 0.6}$ | $-4.5_{\pm 1.1}$ | $0.6_{\pm 0.4}$ | $0.8_{\pm 0.3}$ |
| Simple | $-3.0_{\pm 0.5}$ | $-2.0_{\pm 1.1}$ | $\mathbf{-4.8}_{\pm 0.5}$ | $-2.1_{\pm 0.3}$ |
| V1 stoch. | $\mathbf{-8.8}_{\pm 0.3}$ | $\mathbf{-30.6}_{\pm 0.4}$ | $1.5_{\pm 0.5}$ | $2.6_{\pm 0.1}$ |

to the standard ResNet50. However, this improvement accounted for only a fraction of the total gains of the VOneResNet50 model (Fig. 3 B), demonstrating that there is a substantial supralinear interaction between the V1 features and the neuronal stochasticity. Merely adding V1 stochasticity to the first block of a standard CNN model does not increase robustness to the same degree as the full VOneBlock—the presence of V1 features more than doubles the contribution to perturbation robustness brought by the addition of neuronal stochasticity.

Finally, we sought to determine whether stochasticity during inference is key to defending against attacks. Thus, we analyzed the white box adversarial performance of VOneResNet50 while quenching stochasticity during the adversarial attack (Fig. 4). Remarkably, the majority of improvements in adversarial robustness originate from the neuronal stochasticity during training, indicating that V1 stochasticity induces the downstream layers to learn representations that are more robust to adversarial attacks. This is particularly interesting when contrasted with the ANT$^{3\times3}$+SIN defense, which has noise added to input images during training, but does not learn representations with notably higher robustness to white box adversarial attacks (Table 1 and Supplementary Fig. B.1).

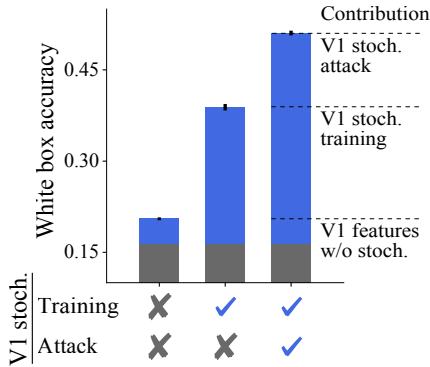

Figure 4: **V1 stochasticity during training leads to more robust learned representations.** White box accuracy (top-1) for VOneResNet50 variant without V1 stochasticity, VOneResNet50 with stochasticity removed during the white box attack and inference, and VOneResNet50. VOneResNet50 without stochasticity during attack/inference maintained over half of the robustness gains. Therefore, V1 stochasticity improves model robustness in two ways: rendering the attack itself less effective and inducing more robust representations downstream of the VOneBlock, the latter accounting for the majority of the gains on our benchmark. Gray bars show the performance of the standard ResNet50. Blue bars show the improvements due to the VOneBlock. Error-bars represent SD (n=3 seeds).

## 5 Discussion

In this work, we demonstrate that adversarial robustness in CNNs is correlated with their ability to explain primate V1 neuronal responses, and that simulating the image processing of primate V1 at the front of standard neural network architectures significantly improves their robustness to image perturbations. Notably, this approach outperformed state-of-the-art defense methods on a large benchmark consisting of adversarial attacks and common image corruptions. Despite not being constructed to this end, any component of the model front-end we removed or modified to be less like primate V1 resulted in less overall robustness, and revealed that different components improve robustness to different perturbation types. Remarkably, we find that simulated V1 stochasticity

interacts synergistically with the V1 model features, and drives the downstream layers to learn representations more robust to adversarial perturbations.

Our approach bears some similarity to the pioneering study by Li et. al. [40], in which a model's representations were regularized to approximate mouse V1, increasing its robustness in the grey-scale CIFAR dataset; however, here we go further in several key ways. First, while the mouse is gaining traction as a model for studying vision, a vast literature has established macaque vision as a quantitatively accurate model of human vision in general and human object recognition in particular [85, 86]. Visual acuity in mice is much lower than in macaques and humans [87, 79], suggesting that vision in the mouse may serve different behavioral functions than in primates. Further, the regularization approach employed by Li et. al. does not allow a clear disambiguation of which aspects of mouse V1 contribute to the improved robustness. Since the components of the VOneBlock proposed here are mappable to the brain, we can dissect the contributions of different neuronal populations in primate V1 to robustness against specific image perturbations. Finally, extending the robustness gains of biologically-constrained CNNs from gray-scale CIFAR to the full ImageNet dataset is a critical step towards real-world, human-level applications.

The gains achieved by VOneNets are substantial, particularly against white box attacks, and have tangible benefits over other defense methods. Though adversarial training still provides the strongest defense for the attack statistics it is trained on, it has significant downsides. Beyond its considerable additional computational cost during training, adversarially trained networks have significantly lower performance on clean images, corrupted images, and images perturbed with attack statistics not seen during training, implying that adversarial training in its current form may not be viable as a general defense method. In contrast, by deploying an architectural change, **VOneNets improve robustness to all adversarial attacks tested and many common image corruptions, and they accomplish this with no additional training overhead**. This also suggests that the architectural gains of the VOneNet could be stacked with other training based defense methods to achieve even greater overall robustness gains.

Relative to current methods, the success of this approach derives from engineering in a better approximation of the architecture of the most well studied primate visual area, combined with task optimization of the remaining free parameters of the downstream architecture [26]. This points to two potentially synergistic avenues for further gains: an even more neurobiologically precise model of V1 (i.e. a better VOneBlock), and an even more neurobiologically precise model of the downstream architecture. For example, one could extend the biological fidelity of the VOneBlock, in the hope that it confers even greater robustness, by including properties such as divisive normalization [48, 49] and contextual modulation [88], to name a few. In addition to V1, the retina and the Lateral Geniculate Nucleus (LGN) also play important roles in pre-processing visual information, only partially captured by the current V1 model, suggesting that extending the work done here to a retina/LGN front-end has potential to better align CNNs with human visual object recognition [89, 90]. In addition, though our initial experiments show that multiple components of our V1 model work together to produce greater robustness, the nature of the relationship between adversarial robustness and explained variance in V1 responses is far from resolved. In particular, as the VOneNet with and without stochasticity achieve very similar V1 explained variances but have markedly different levels of adversarial robustness, it is clear that much theoretical work remains in better understanding both when and why matching biology leads to more robust computer vision models.

While neuroscience has recently seen a huge influx of new neurally-mechanistic models and tools drawn from machine learning [91, 92], the most recent advances in machine learning and computer vision have been driven mostly by the widespread availability of computational resources and very large labeled datasets [93], rather than by an understanding of the relevant brain mechanisms. Under the belief that biological intelligence still has a lot of untapped potential to contribute, a number of researchers have been pushing for more neuroscience-inspired machine learning algorithms [37, 42, 32]. The work presented here shows that this aspiration can become reality—the models presented here, drawn directly from primate neurobiology, indeed require less training to achieve more human-like behavior. This is one turn of a new virtuous circle, wherein neuroscience and artificial intelligence each feed into and reinforce the understanding and ability of the other.

## Broader Impact

From a technological perspective, the ethical implications of our work are largely aligned with those of computer vision in general. While there is undoubtedly potential for malicious and abusive uses of computer vision, particularly in the form of discrimination or invasion of privacy, we believe that our work will aid in the production of more robust and intuitive behavior of computer vision algorithms. As CNNs are deployed in real-world situations, it is critical that they behave with the same level of stability as their human counterparts. In particular, they should at the very least not be confused by changes in input statistics that do not confuse humans. We believe that this work will help to bridge that gap. Furthermore, while algorithms are often thought to be impartial or unbiased, much research has shown that data driven models like current CNNs are often even more biased than humans, implicitly keying in on and amplifying stereotypes. For this reason, making new CNNs that behave more like humans may actually reduce, or at least make more intuitive, their implicit biases. Unfortunately, we note that even with our work, these issues are not resolved, yet. While we developed a more neurobiologically-constrained algorithm, it comes nowhere close to human-like behaviour in the wide range of circumstances experienced in the real world. Finally, from the perspective of neuroscience, we think that this work introduces a more accurate model of the primate visual system. Ultimately, better models contribute to a better mechanistic understanding of how the brain works, and how to intervene in the case of illness or disease states. We think that our model contributes a stronger foundation for understanding the brain and building novel medical technology such as neural implants for restoring vision in people with impairments.

## Acknowledgments and Disclosure of Funding

We thank J. Anthony Movshon and Corey M. Ziemba for access to the V1 neural dataset, MIT-IBM Satori for providing the compute necessary for this project, John Cohn for technical support, Sijia Liu and Wieland Brendel for advice on adversarial attacks, and Adam Marblestone for insightful discussion. This work was supported by the PhRMA Foundation Postdoctoral Fellowship in Informatics (T.M), the Semiconductor Research Corporation (SRC) and DARPA (J.D., M.S., J.J.D.), the Massachusetts Institute of Technology Shoemaker Fellowship (M.S.), Office of Naval Research grant MURI-114407 (J.J.D.), the Simons Foundation grant SCGB-542965 (J.J.D.), the MIT-IBM Watson AI Lab grant W1771646 (J.J.D.).

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
