[Supplementary Material]

# Supplementary Material

## A    Neural Explained Variance

We evaluated how well responses to given images in candidate CNNs explain responses of single V1 neurons to the same images using a standard neural predictivity methodology based on partial least square (PLS) regression [31, 32]. We used a neural dataset obtained by extracellular recordings from 102 single-units while presenting 450 different images for 100ms, 20 times each, spanning $4\deg$ of the visual space. Results from this dataset were originally published in a study comparing V1 and V2 responses to naturalistic textures [77]. To avoid over-fitting, for each CNN, we first chose its best V1 layer on a subset of the dataset (n=135 images) and reported the final neural explained variance calculated for the chosen layer on the remaining images of the dataset (n=315 images). Model units were mapped to each V1 neuron linearly using a PLS regression model with 25 components. The mapping was performed for each neuron using 90% of the image responses in the respective dataset and tested on the remaining held-out 10% in a 10-fold cross-validation strategy. We reported neural explained variance values normalized by the neuronal internal consistency. We calculated V1 neural explained variance for a large pool of CNNs (n=30 models).

When using CNNs in a computer vision task, such as object recognition in ImageNet, their field-of-view (in degrees) is not defined since the only relevant input property is the image resolution (in pixels). However, when using a CNN to model the ventral stream, the visual spatial extent of the model's input is of key importance to ensure that it is correctly mapped to the data it is trying to explain. We assigned a spatial extent of $8\deg$ to all models since this value has been previously used in studies benchmarking CNNs as models of the primate ventral stream [31, 94], and is consistent with the results in Cadena et. al. [30].

# B   Image Perturbations

## B.1   White Box Adversarial Attacks

For performing white box adversarial attacks, we used untargeted projected gradient descent (PGD) [17] (also referred to as the Basic Iterative Method [95]) with $L_\infty$, $L_2$, and $L_1$ norm constraints. Given an image $x$, This method uses the gradient of the loss to iteratively construct an adversarial image $x_{adv}$ which maximizes the model loss within an $L_p$ bound around $x$. Formally, in the case of an $L_\infty$ constraint, PGD iteratively computes $x_{adv}$ as

$$x^{t+1} = Proj_{x+\mathcal{S}}(x^t + \alpha sgn(\nabla_{x^t} L(\theta, x^t, y)))$$

where $x$ is the original image, and the $Proj$ operator ensures the final computed adversarial image $x_{adv}$ is constrained to the space $x + \mathcal{S}$, here the $L_\infty$ ball around $x$. In the case of $L_1$ or $L_2$ constraints, at each iteration $\nabla_{x^t} L(\theta, x^t, y)$ is scaled to have an $L_1$ or $L_2$ norm of $\alpha$, and the $Proj$ operator ensures the final $x_{adv}$ is within an $L_1$ or $L_2$ norm ball around $x$.

For our white box benchmarks, we used $\|\delta\|_\infty \in [1/1020, 1/255]$, $\|\delta\|_2 \in [0.15, 0.6]$, and $\|\delta\|_1 \in [40, 160]$ constraints where $\delta = x - x_{adv}$, for a total of six different attack variants, with two strengths under each norm constraint. We arrived at these strengths by calibrating the lower strength attack for each norm to give a reasonable degree of variance in normally trained CNNs. Because normally trained CNNs are extremely sensitive to adversarial perturbations, this resulted in perturbations well below the level of perceptibility, and in most cases, particularly with the $L_\infty$ constraint, perturbations that would not even be rendered differently by a monitor for display. We set the high attack strength at 4 times the low value, which brought standard ImageNet trained CNNs to nearly chance performance. Still, even this higher perturbation strength remained practically imperceptible (Fig. B.3 left). Each perturbation was computed with 64 PGD iterations and a step size $\alpha = \epsilon/32$, where $\epsilon = \|\delta\|_p$, for the same 5000 images from the ImageNet validation set, on a per model basis, and final top-1 accuracy was reported. The Adversarial Robustness Toolkit [96] was used for computing the attacks.

While adversarial formulations using Lagrangian relaxation like the C&W attack [7] and the EAD attack [8] are known to be more effective than PGD for finding minimal perturbations under $L_2$ and $L_1$ constraints, these formulations involve computationally expensive line searches, often requiring thousands of iterations to converge, making them computationally difficult to scale to 5000 ImageNet samples for a large number of models. Furthermore, because some of the models we tested operate with stochasticity during inference, we found that other state-of-the-art attacks formulated to efficiently find minimal perturbation distances [9, 10] were generally difficult to tune, as their success becomes stochastic as they near the decision boundary. Thus we proceeded with the PGD formulation as we found it to be the most reliable, computationally tractable, and conceptually simple to follow. We share our model and weights, and encourage anyone interested to attack our model to verify our results.

Although the stochastic models were implemented with the reparameterization trick for random sampling, which leaves the gradients intact [83], we performed several sanity checks to make sure that the white box attacks were effectively implemented. Following the advice of Athalye et. al. [68] and Carlini et. al. [84], we observed that in all of the white box attacks, increasing the attack strength decreased model performance, eventually bringing accuracy to zero (Fig. B.1), and additionally that for any given constraint and attack strength, increasing PGD iterations from one to many increased attack effectiveness (Fig. B.2). Furthermore, to be extra cautious, we followed the method Athalye et. al. [68] used to break Stochastic Activation Pruning [19] and replaced $\nabla_x f(x)$ with $\sum_{i=1}^{k} \nabla_x f(x)$, effectively performing Monte Carlo (MC) sampling of the loss gradient at every PGD iteration. While with high perturbation strength even $k = 1$ was sufficient to bring our model's accuracy to zero, we find that $k = 10$ generally achieved higher attack success at intermediate and large perturbation strengths (Table B.1), and thus we used this approach for all attacks on stochastic models.

Figure B.1: **VOneResNet50 improves robustness to white box attacks in a wide range of perturbation strengths.** White box accuracy perturbation strength curves for PGD attacks with constraints $L_\infty$, $L_2$ and $L_1$ for ResNet50, VOneResNet50, $\text{AT}_{L_\infty}$, and $\text{ANT}^{3\times3}$+SIN. Adding the VOneBlock to ResNet50 consistently improves robustness under all constraints and attack strength—VOneResNet50 can withstand perturbations roughly 4 times higher than the standard ResNet50 for the same performance level. Remarkably, VOneResNet50 outperformed the adversarially trained model ($\text{AT}_{L_\infty}$) for a considerable range of perturbation strengths, particularly with the $L_1$ constraint. Adversarial noise training combined with Stylized ImageNet ($\text{ANT}^{3\times3}$+SIN) has virtually no improvement in robustness to white box attacks. In all white box attacks and for all models, increasing the attack strength decreased performance, eventually bringing accuracy to zero.

Figure B.2: **White box attack effectiveness converges for large number of PGD iteration steps.** White box accuracy iteration curves for PGD attacks with $\|\delta\|_\infty = 1/255$, $\|\delta\|_2 = 0.6$, $\|\delta\|_1 = 160$ constraints. Increasing the number of PGD iteration steps makes the attack more effective, generally converging after 32 iterations.

Table B.1: **Improving the effectiveness of white box attacks with Monte Carlo sampling**. White box attack accuracy (top-1) under different norm constraints and white box overall mean using standard PGD and PGD with Monte Carlo sampling of the loss gradients (PGD-MC, $k = 10$) on VOneResNet50 for 64 PGD iteration steps. For all stochastic models we used PGD with Monte Carlo sampling in the white box attacks throughout the paper. Values are mean and SD (n=3 seeds).

| | White box | PGD-$L_1$ | | PGD-$L_2$ | | PGD-$L_\infty$ | |
| | Mean | 40 | 160 | 0.15 | 0.60 | 1/1020 | 1/255 |
| Attack type | [%] | [%] | [%] | [%] | [%] | [%] | [%] |
|---|---|---|---|---|---|---|---|
| Standard PGD | $52.8_{\pm0.2}$ | $64.4_{\pm0.4}$ | $42.2_{\pm0.1}$ | $65.2_{\pm0.3}$ | $45.3_{\pm0.4}$ | $63.4_{\pm0.4}$ | $36.4_{\pm0.4}$ |
| PGD-MC ($k$=10) | $51.1_{\pm0.4}$ | $64.2_{\pm0.5}$ | $40.4_{\pm0.5}$ | $65.3_{\pm0.4}$ | $44.3_{\pm0.3}$ | $61.9_{\pm0.4}$ | $30.3_{\pm0.5}$ |

Although VOneResNet50 was not designed to explicitly improve robustness to white box attacks, it offers substantial gains over the standard ResNet50 model (Table 1). A closer inspection of the white box accuracy perturbation strength curves (Fig. B.1) reveals that VOneResNet50 can withstand perturbations roughly 4 times higher than the standard ResNet50 for the same performance level (blue curve is shifted to the right of the black curve by a factor of 4). To help visualize how this difference translates into actual images, we show in Fig. B.3 examples of white box adversarial images for both ResNet50 and VOneResNe50 for the perturbation strengths that bring the respective models to nearly chance performance.

Figure B.3: **Comparison of adversarial images for ResNet50 and VOneResNet50.** Visualization of white box attacks under the 3 different norm constraints ($L_\infty$, $L_2$, and $L_1$) for the perturbation strength that brings the model to nearly chance performance (Fig. B.1). Left, ResNet50; right, VOneResNet50. For each attack, the perturbation is shown on the left, scaled for visualization, and the perturbed image appears next to it. Perturbation strengths and classes of the adversarial images are shown over the images.

## B.2 Additional attack controls

The field of adversarial robustness is littered with examples of proposed defense methods that were quickly defeated with some minor attack optimizations or adaptations [68, 84, 97]. For this reason, we took great care to verify our claims of improved robustness of the VOneNets. In addition to the implementation details and controls described in the previous section, we validated our results by further optimizing our attacks on both the standard stochastic VOneResNet50 and the VOneResNet50 with stochasticity removed during the attack and inference.

First, on the standard stochastic VOneResNet50, we ensured that our attack hyper-parameters, including step-size ($\alpha$), number of gradient samples ($k$), and PGD iterations, are sufficient for evaluating the network's robustness. For PGD with $\|\delta\|_\infty = 1/255$ with 64 iterations, we performed a large grid search over $\alpha$ and $k$, observing only a marginal increase in the attack effectiveness (Fig. B.4). For the most effective step size ($\alpha = \epsilon/8$), increasing the number of gradient samples beyond $k = 16$ no longer decreases accuracy under the attack. At $k = 128$, VOneResNet50 accuracy is only reduced from 29.15% to 26.0%, remaining a large margin above ResNet50 accuracy of 0.8%. Additionally, increasing the number of PGD iterations all the way up to 1000 (with $\alpha = \epsilon/8$ and $k = 16$), only further reduced performance by 0.5%. Finally, using 5-10 random restarts did not improve the attack beyond the most effective $k$ and $\alpha$ settings.

Figure B.4: **Further attack optimization does not overturn results.** White box accuracy for PGD with $\|\delta\|_\infty = 1/255$, 64 iterations on 500 images. Large grid search over step size and number of gradient samples only marginally improves attack effectiveness. Further attack optimization only marginally reduces model performance. Lines represent mean and shaded areas SD (n=3 seeds).

Second, while boundary attacks are not ideal against the standard stochastic VOneResNet50 due to the shifting boundary line, we investigated VOneResNet50 without stochasticity during attack/inference using the Brendel and Bethge (B&B) attack [10], a powerful white box adversary that aims to find minimal adversarial perturbations along the decision boundary of the network. We reasoned that if the B&B attack offers similar results to those observed using PGD, this is a good sanity check that our observed robustness is reliable, and not a result of masked gradients around the input or a quirk of PGD. Using the FoolBox [98] implementation of the $L_\infty$ B&B attack initialized with 500 successful adversarial images generated using PGD $\|\delta\|_\infty = 8/255$, we see nearly identical scores as compared to standard PGD with optimal hyper-parameters $\|\delta\|_\infty = 1/1020$ and PGD $\|\delta\|_\infty = 1/255$.

We conclude that these additional controls do not qualitatively change our reported results and provide further evidence that the observed improved robustness of the VOneNets is real and not an artifact.

## B.3  Common Image Corruptions

For evaluating model robustness to common image corruptions, we used ImageNet-C, a dataset publicly available at `https://github.com/hendrycks/robustness` [13]. It consists of 15 different types of corruptions, each at five levels of severity for a total of 75 different perturbations (Fig. B.5). The corruption types fall into four categories: noise, blur, weather, and digital effects. Noise includes Gaussian noise, shot noise, and impulse noise; blur includes motion blur, defocus blur, zoom blur, and glass blur; weather includes snow, frost, fog, and brightness; digital effects includes JPEG compression, elastic transformations, contrast, and pixelation. For every corruption type and at every severity, we tested on all 50,000 images from the ImageNet validation set. We note that these images have been JPEG compressed by the original authors for sharing, which is known to have minor but noticeable effect on final network performance.

Figure B.5: **Common image corruptions with intermediate severity.** Visualization of all 15 common image corruption types evaluated at severity = 3. First row, original image, followed by the noise corruptions; second row, blur corruptions; third row, weather corruptions; fourth row, digital corruptions.

Figure B.6: **VOneResNet50 improves robustness to noise, blur and digital corruptions.** Corruption accuracy severity curves for the four categories of common image corruptions: noise, blur, weather, and digital. Adding the VOneBlock to ResNet50 improves robustness to all corruption categories except weather. The adversarially trained model ($\text{AT}_{L_\infty}$) is considerably worse at common image corruptions. Adversarial noise combined with Stylized ImageNet training ($\text{ANT}^{3\times3}$+SIN) is consistently more robust to all corruption categories. For ResNet50 and VOneResNet50, error-bars represent SD (n=3 seeds).

Table B.2: **Detailed common image corruption results.** Adding the VOneBlock to ResNet50 improves robustness to several of the corruption types. However, for others, particularly fog and contrast, robustness decreases. Interestingly, these are the corruption types that the adversarially trained model ($\text{AT}_{L_\infty}$) also does the worst. Adversarial noise combined with Stylized ImageNet training ($\text{ANT}^{3\times3}$+SIN) is the most robust model for most of the corruption types. For ResNet50 and VOneResNet50, values are mean and SD (n=3 seeds).

| | Noise | | | Blur | | | |
|---|---|---|---|---|---|---|---|
| Model | Gaussian [%] | Shot [%] | Impulse [%] | Defocus [%] | Glass [%] | Motion [%] | Zoom [%] |
| ResNet50 [4] | $29.6_{\pm1.1}$ | $27.9_{\pm0.9}$ | $24.5_{\pm1.2}$ | $38.8_{\pm0.7}$ | $26.1_{\pm0.6}$ | $37.9_{\pm0.4}$ | $34.5_{\pm0.7}$ |
| $\text{AT}_{L_\infty}$ [76] | 23.6 | 22.3 | 14.8 | 24.4 | 34.3 | 33.2 | 34.9 |
| $\text{ANT}^{3\times3}$+SIN [59] | **61.6** | **60.7** | **59.9** | **41.5** | **38.6** | **46.0** | **37.3** |
| VOneResNet50 | $34.6_{\pm1.4}$ | $33.4_{\pm1.2}$ | $31.9_{\pm1.5}$ | $37.8_{\pm0.1}$ | $35.7_{\pm0.1}$ | $37.4_{\pm0.2}$ | $34.0_{\pm0.2}$ |

| | Weather | | | | Digital | | | |
|---|---|---|---|---|---|---|---|---|
| Model | Snow [%] | Frost [%] | Fog [%] | Bright. [%] | Contrast [%] | Elastic [%] | Pixelate [%] | JPEG [%] |
| ResNet50 | $30.1_{\pm0.5}$ | $36.7_{\pm0.4}$ | $43.6_{\pm0.8}$ | $66.8_{\pm0.2}$ | $38.8_{\pm0.5}$ | $44.8_{\pm0.3}$ | $47.0_{\pm2.7}$ | $55.1_{\pm1.0}$ |
| $\text{AT}_{L_\infty}$ | 29.5 | 31.2 | 8.4 | 54.1 | 10.2 | 48.4 | 56.3 | 58.7 |
| $\text{ANT}^{3\times3}$+SIN | **47.7** | **52.2** | **54.9** | **68.2** | **49.3** | **52.7** | 58.9 | 59.3 |
| VOneResNet50 | $25.2_{\pm0.5}$ | $36.8_{\pm0.4}$ | $30.1_{\pm0.5}$ | $62.4_{\pm0.3}$ | $28.5_{\pm0.5}$ | $48.7_{\pm0.1}$ | $\mathbf{63.3}_{\pm0.2}$ | $\mathbf{61.0}_{\pm0.3}$ |

## B.4 Other defense methods

We compared the performance of VOneResNet50 in the described computer vision benchmarks with two training-based defense methods using the ResNet50 architecture: adversarial training with a $\|\delta\|_\infty = 4/255$ constraint ($\text{AT}_{L_\infty}$) [76], and adversarial noise with Stylized ImageNet training ($\text{ANT}^{3\times3}$+SIN) [59].

The $\text{AT}_{L_\infty}$ model was downloaded from the publicly available adversarially trained models at `https://github.com/MadryLab/robustness` [76]. This model was trained following the adversarial training paradigm of Madry et. al. [17], to solve the min-max problem,

$$\min_\theta \rho(\theta), \text{ where } \rho(\theta) = \mathbb{E}_{(x,y)\sim\mathcal{D}} \left[ \max_{\delta\in\mathcal{S}} L(\theta, x + \delta, y) \right] \tag{1}$$

Here, the goal is to learn parameters $\theta$ minimizing the loss $L$ for training images $x$ and labels $y$ drawn from $\mathcal{D}$, while perturbing $x$ with $\delta \in \mathcal{S}$ to maximally increase the loss. In this case, $\mathcal{D}$ is ImageNet, and PGD with $\|\delta\|_\infty = 4/255$ is used to find the perturbation maximizing the loss for a given $(x, y)$ pair. The $\|\delta\|_\infty = 4/255$ constraint model was selected to compare against because it had the best performance on our conglomerate benchmark of adversarial attacks and common image corruptions.

The $\text{ANT}^{3\times3}$+SIN model was obtained from `https://github.com/bethgelab/game-of-noise`. Recently, Rusak et. al. showed that training a ResNet50 model with several types of additive input noise improved the robustness to common image corruptions [59]. Inspired by this observation, the authors decided to train a model while optimizing a noise distribution that maximally confuses it. Similarly to standard adversarial training, this results in solving a min-max problem,

$$\min_\theta \max_\phi \mathbb{E}_{(x,y)\sim\mathcal{D}}\mathbb{E}_{\delta\sim p_\phi(\delta)} \left[ L(\theta, x + \delta, y) \right] \tag{2}$$

where $p_\phi(\delta)$ is the maximally confusing noise distribution. The main difference to regular adversarial training in Equation 1, is that while in the former $\delta$ is optimized directly, here $\delta$ is found by optimizing a constrained distribution with local spatial correlations. Complementing the adversarial noise training with Stylized ImageNet training [25] produced the model with the current best robustness in the ImageNet-C benchmark as a standalone method.

## C  VOneNet implementation details

### C.1  Convolutional layer

The convolutional layer of the VOneBlock is a mathematically parameterized Gabor Filter Bank (GFB). We set the stride of the GFB to be four, originating a 56×56 spatial map of activations. Since the number of channels in most CNNs' first convolution is relatively small (64 in the architectures adapted), we used a larger number in the VOneBlock so that the Gabors would cover the large parameter space and better approximate primate V1. We set the main VOneNet models to contain 512 channels equally split between simple and complex cells (see Fig. C.2 A for an analysis of how the number of channels affects performance). Each channel in the GFB convolves a single color channel from the input image.

The Gabor function consists of a two-dimensional grating with a Gaussian envelope and is described by the following equation:

$$G_{\theta,f,\phi,n_x,n_y}(x,y) = \frac{1}{2\pi\sigma_x\sigma_y} \exp\left[-0.5(x_{rot}^2/\sigma_x^2 + y_{rot}^2/\sigma_y^2)\right]\cos\left(2\pi f + \phi\right) \qquad (3)$$

where

$$\begin{aligned} x_{rot} &= x\cos(\theta) + y\sin(\theta) \\ y_{rot} &= -x\sin(\theta) + y\cos(\theta) \end{aligned} \qquad (4) \qquad\qquad \begin{aligned} \sigma_x &= \frac{n_x}{f} \\ \sigma_y &= \frac{n_y}{f} \end{aligned} \qquad (5)$$

$x_{rot}$ and $y_{rot}$ are the orthogonal and parallel orientations relative to the grating, $\theta$ is the angle of the grating orientation, $f$ is the spatial frequency of the grating, $\phi$ is the phase of the grating relative to the Gaussian envelope, and $\sigma_x$ and $\sigma_y$ are the standard deviations of the Gaussian envelope orthogonal and parallel to the grating, which can be defined as multiples ($n_x$ and $n_y$) of the grating cycle (inverse of the frequency).

Although the Gabor filter greatly reduces the number of parameters defining the linear spatial component of V1 neurons, it still has five parameters per channel. Fortunately, there is a vast literature in neurophysiology with detailed characterizations of primate V1 response properties which can be used to constrain these parameters. To instantiate a VOneBlock with $C_{V1}$ channels, we sampled $C_{V1}$ values for each of the parameters according to an empirically constrained distribution (Table C.1). Due to the resolution of the input images, we limited the ranges of spatial frequencies ($f < 5.6$cpd) and number of cycles ($n > 0.1$).

Table C.1: **Empirical distributions used to sample the GFB parameters in the VOneBlock**.

| Parameter | Reference | |
|---|---|---|
| $\theta$ | De Valois et. at. [78] | Fig. 5: Orientation tuning (foveal) |
| $f$ | De Valois et. at. [79] | Fig. 6: Peak spatial freq. (X cells foveal) |
| $\phi$ | - | Uniform distribution $\in [0, 360]$ deg |
| $n_x, n_y$ | Ringach 2002 [80] | Fig. 4: $(n_x, n_y)$ joint distribution |

Critical to the correct implementation of the biologically-constrained parameters of the GFB is the choice of the model's field of view in degrees. As previously mentioned in Section A, we used 8deg as the input spatial extent for all CNN models. It's important to note that the set of spatial filters in the VOneBlock's GFB differs considerably from those in the first convolution in most CNNs (Fig. C.1). While standard CNNs learn filters that resemble Gabors in their input layer [1, 99, 100], due the limited span of their kernels, they do not vary significantly in size and spatial frequency. V1 neurons, on the other hand, are known to exhibit a wide range of receptive field properties. This phenomena is captured in the VOneBlock with the spatial frequencies and receptive field sizes of Gabors ranging more than one order of magnitude. Due to this high variability, we set the convolution kernel to be $25 \times 25$px, which is considerably larger than those in standard CNNs (Fig. C.1).

Figure C.1: **Comparison between filters in the first convolution of standard CNNs and the VOneBlock.** Example filters from the first convolution in the VOneBlock, ResNet50, CORnet-S, and AlexNet (from left to right). VOneBlock filters are all parameterized as Gabors, varying considerably in size and spatial frequency. Standard CNNs have some filters with shapes other than Gabors, particularly center-surround, and are limited in size by their small kernel. Kernel sizes are 25px, 7px, 7px, and 11px for VOneBlock, ResNet50, CORnet-S, and AlexNet, respectively.

## C.2 Nonlinear layer

VOneBlock's nonlinear layer has two different nonlinearities that are applied to each channel depending on its cell type: a rectified linear transformation for simple cells (6), and the spectral power of a quadrature phase-pair (7) for complex cells:

$$S^{nl}_{\theta,f,\phi,n_x,n_y} = \begin{cases} S^{l}_{\theta,f,\phi,n_x,n_y}, & \text{if } S^{l}_{\theta,f,\phi,n_x,n_y} > 0 \\ 0, & \text{otherwise} \end{cases} \tag{6}$$

$$C^{nl}_{\theta,f,\phi,n_x,n_y} = \frac{1}{\sqrt{2}} \sqrt{(C^{l}_{\theta,f,\phi,n_x,n_y})^2 + (C^{l}_{\theta,f,\phi+\pi/2,n_x,n_y})^2} \tag{7}$$

where $S^{l}_{...}$ and $S^{nl}_{...}$ are the linear and nonlinear responses of a simple neuron and $C^{l}_{...}$ and $C^{nl}_{...}$ are the same for a complex neuron.

## C.3 Neuronal stochasticity generator

In awake monkeys, spike trains of V1 neurons are approximately Poisson, i.e. the variance and mean of spike counts, in a given time-window, over a set of repetitions are roughly the same [82]. We incorporated stochasticity into the VOneBlock to emulate this property of neuronal responses. Since the Poisson distribution is not continuous, it breaks the gradients in a white box attack giving a false sense of robustness [68]. In order to avoid this situation and facilitate the evaluation of the model's real robustness, our neuronal stochasticity generator as implemented uses a continuous, second-order approximation of Poisson noise by adding Gaussian noise with variance equal to the activation:

$$R^s \sim \mathcal{N}(\mu = R^{ns}, \sigma^2 = R^{ns}) \tag{8}$$

where $R^{ns}$ and $R^s$ are the non-stochastic and stochastic responses of a neuron.

To approximate the same levels of neuronal stochasticity of primate V1 neurons, it is critical that the VOneBlock activations are on the same range as the V1 neuronal responses (number of spike counts in a given time-window). Thus, we applied an affine transformation to the activations so that both the mean stimulus response and the mean baseline activity are the same as those of a population of V1 neurons (Table C.2 shows the mean responses and spontaneous activity of V1 neurons measured in different time-windows).

Table C.2: **Mean stimulus responses and spontaneous activity of a population of V1 neurons for different time-windows**.

| Time-window [ms] | Stimulus response [spikes] | Spontaneous activity [spikes] |
|---|---|---|
| 25 | 0.324 | 0.073 |
| 50 | 0.655 | 0.145 |
| 100 | 1.05 | 0.29 |
| 200 | 1.38 | 0.58 |

Since the exact time-window that responses in V1 are integrated during visual perception is still an open question, we considered a time-window of 25ms (Fig. C.2 B). In order to keep the outputs of the VOneBlock on a range that does not deviate considerably from the typical range of activations in CNNs, so that the model can be efficiently trained using standard training parameters, we applied the inverse transformation to scale the outputs back to their original range after the stochastic layer.

## C.4 Hyper-parameters and other design choices

We developed VOneNets from three different standard CNN architectures: ResNet50, CORnet-S, and AlexNet. As previously mentioned, we replaced the first block of each architecture by the VOneBlock and the transition layer to create the respective VOneNet. Table C.3 contains the layers that were removed for each base architecture. Except CORnet-S, the layers removed only contained a single

convolution, nonlinearity and maxpool. Since CORnet-S already had a pre-committed set of layers to V1, we replaced them by the VOneBlock. The torchvision implementation of AlexNet contains a combined stride of eight in the removed layers (four in the convolution and two in the maxpool), followed by a stride of one in the second convolution (outside of the removed layers). In order to more easily adapt it to a VOneBlock with a stride of four, we slightly adapted AlexNet's architecture so that it had strides of two in these three layers (first convolution, first maxpool, and second convolution). The results shown in Fig. 2 were obtained using this modified architecture.

Table C.3: **Layers removed from each base CNN architecture for creating the VOneNets.**

| Architecture | Layers removed |
|---|---|
| AlexNet | Conv/ReLU/MaxPool |
| ResNet50 | Conv/Batch-Norm/ReLU/MaxPool |
| CORnet-S | Conv/Batch-Norm/ReLU/MaxPool/Conv/Batch-Norm/ReLU |

The VOneNet architecture was designed to have as few hyper-parameters as possible. When possible these were either constrained by the base CNN architecture or by neurobiological data: most parameters of the GFB were instantiated by sampling from neuronal distributions of primate V1 neurons; the kernel size was set to $25\times25$px to capture the high variability in Gabor sizes; like all other CNN models, the spatial extent of the field of view was set to 8deg (Section A). Nevertheless, there were two hyper-parameters where the choice was rather arbitrary: the number of channels in the VOneBlock, and the time-window for integrating V1 responses to scale the activations prior to the stochasticity generator.

Since primate V1 has neurons tuned to a wide range of spatial frequencies, for each orientation and at any given location of the visual space [101], we expected that a large number of channels would be required to cover all of the combinations of Gabor parameters. For the VOneResNet50 architecture, we varied the number of channels in the VOneBlock between 64 and 1024, equally split between simple and complex cells, and measured the performance of the several models after training (Fig. C.2 A). Clean ImageNet and common image corruption performance improved with channel number. Remarkably, the opposite happened for white box accuracy, with variants of VOneResNet50 with 64 and 128 channels achieving the highest robustness to white box attacks. This result is particularly interesting since it had been shown that increasing the number of channels in all layers of a CNN improves robustness to white box attacks in both standard and adversarially trained models [17]. Therefore, this shows that the improvement in white box robustness of VOneNets, when compared to their base models, cannot be attributed to the higher number of channels. For the main model, we set the channel number to be 512 as it offered a good compromise between the different benchmarks. Regarding the time-window for integrating V1 responses, we observed small effects on the models' performance when varying it between 25 and 200ms. We chose 25ms for the main model due to a small trend for higher robustness to white box attacks with shorter time-windows.

Figure C.2: **Exploration of VOneBlock's hyper-parameters in the VOneResNet50 architecture. A** Performance of different VOneResNet50 models with varying number of channels in the VOneBlock. **B** Same as in **A** but varying time-window for integrating neuronal activity. Error-bars show SD (n=3 seeds). Red arrows represent the values used in the main model.

## C.5 Training details

We used PyTorch 0.4.1 and trained the model using ImageNet 2012 [34]. Images were preprocessed (1) for training—random crop to $224 \times 224$ pixels and random flipping left and right; (2) for validation—central crop to $224 \times 224$ pixels. Preprocessing was followed by normalization—subtraction and division by [0.5, 0.5, 0.5]. We used a batch size of 256 images and trained either on 2 GPUs (NVIDIA Titan X / GeForce 1080Ti) or 1 GPU (QuadroRTX6000 or V100) for 70 epochs. We use step learning rate scheduling: 0.1 starting learning rate, divided by 10 every 20 epochs. For optimization, we use Stochastic Gradient Descent with a weight decay 0.0001, momentum 0.9, and a cross-entropy loss between image labels and model predictions (logits).

Model weights and code are available at `https://github.com/dicarlolab/vonenet`.

# D VOneBlock variant details

To investigate which components of the VOneBlock are responsible for the increased robustness, we created six variants of the VOneBlock by removing or modifying one of its parts. In all cases, we trained from scratch a corresponding variant of VOneResNet50 with the modified VOneBlock. All other properties of the model and components of the VOneBlock were left unmodified. Here we describe in more detail each of these variants (the name of the variant refers to the component removed):

- *V1 distributions:* the GFB parameters were sampled from uniform distributions with the same domain as the empirical distributions used in the default VOneBlock (Table C.1).
- *Low SF:* when sampling the spatial frequencies for the GFB, neurons with low peak spatial frequencies ($f < 2$cpd) were removed from the empirical distribution [79].
- *High SF:* similar to the previous variant but removing neurons with high peak spatial frequencies ($f > 2$cpd).
- *Complex NL:* complex cell channels were removed from the VOneBlock and replaced by simple cell channels.
- *Simple NL:* simple cell channels were removed from the VOneBlock and replaced by complex cell channels.
- *V1 stochasticity:* V1 stochasticity generator was removed from the VOneBlock.

While all of the variants resulted in worse overall mean performance than the unmodified VOneResNet50 (Table 2), some improved specific benchmarks: the variant without high SF had higher accuracy under noise and blur corruptions (Table D.1), the variant without V1 stochasticity had higher accuracy to clean images and images with noise and weather corruptions (Tables 2 and D.1), and the variant without V1 distributions had higher accuracy to white box attacks with high $L_\infty$ perturbations (Table D.2).

Table D.1: **Difference between the common image corruption accuracies of the several VOneResNet50 variants and the unmodified VOneResNet50**. Removal of simple cells reduced accuracy to all common image corruption types. Removal of some components of the VOneBlock improved accuracy in specific common image corruption: removing high SF consistently improved performance for noise and blur corruptions, and removing V1 stochasticity improved performance for noise and weather corruptions.

| Component removed | Noise | | | Blur | | | |
| --- | --- | --- | --- | --- | --- | --- | --- |
| | Gaussian [$\Delta\%$] | Shot [$\Delta\%$] | Impulse [$\Delta\%$] | Defocus [$\Delta\%$] | Glass [$\Delta\%$] | Motion [$\Delta\%$] | Zoom [$\Delta\%$] |
| V1 dist. | -2.8$_{\pm0.9}$ | -2.4$_{\pm0.7}$ | -2.9$_{\pm1.1}$ | -1.6$_{\pm0.1}$ | 0.4$_{\pm0.2}$ | -0.8$_{\pm0.3}$ | -0.2$_{\pm0.5}$ |
| Low SF | -1.4$_{\pm0.5}$ | -1.0$_{\pm0.2}$ | -1.5$_{\pm0.5}$ | -0.6$_{\pm0.6}$ | -1.1$_{\pm0.2}$ | -1.2$_{\pm0.4}$ | -0.5$_{\pm0.7}$ |
| High SF | **6.8**$_{\pm1.3}$ | **7.3**$_{\pm1.1}$ | **7.0**$_{\pm1.3}$ | **5.6**$_{\pm0.3}$ | **11.5**$_{\pm0.3}$ | **3.7**$_{\pm0.0}$ | **6.1**$_{\pm0.1}$ |
| Complex | 1.0$_{\pm1.6}$ | 1.2$_{\pm1.3}$ | 0.7$_{\pm1.8}$ | 1.2$_{\pm0.2}$ | -1.5$_{\pm0.1}$ | 0.2$_{\pm0.8}$ | 0.4$_{\pm1.1}$ |
| Simple | -6.1$_{\pm1.6}$ | -5.4$_{\pm1.5}$ | -6.4$_{\pm1.9}$ | -4.1$_{\pm0.4}$ | -4.2$_{\pm0.4}$ | -4.9$_{\pm0.5}$ | -5.4$_{\pm0.3}$ |
| V1 stoch. | 2.2$_{\pm2.5}$ | 2.2$_{\pm2.5}$ | 2.1$_{\pm2.9}$ | 3.3$_{\pm0.1}$ | -5.5$_{\pm0.1}$ | 0.9$_{\pm0.7}$ | 0.9$_{\pm0.1}$ |

| Component removed | Weather | | | | Digital | | | |
| --- | --- | --- | --- | --- | --- | --- | --- | --- |
| | Snow [$\Delta\%$] | Frost [$\Delta\%$] | Fog [$\Delta\%$] | Bright. [$\Delta\%$] | Contrast [$\Delta\%$] | Elastic [$\Delta\%$] | Pixelate [$\Delta\%$] | JPEG [$\Delta\%$] |
| V1 dist. | -0.2$_{\pm0.6}$ | -0.9$_{\pm0.6}$ | -1.9$_{\pm1.1}$ | -0.8$_{\pm0.5}$ | -1.4$_{\pm0.9}$ | 0.3$_{\pm0.4}$ | -1.7$_{\pm0.3}$ | **0.8**$_{\pm0.5}$ |
| Low SF | -0.4$_{\pm0.3}$ | -0.6$_{\pm0.4}$ | 0.9$_{\pm0.3}$ | -0.2$_{\pm0.2}$ | -0.4$_{\pm0.5}$ | -0.9$_{\pm0.5}$ | -1.3$_{\pm0.2}$ | -0.5$_{\pm0.4}$ |
| High SF | 1.4$_{\pm1.0}$ | -0.7$_{\pm0.7}$ | -9.5$_{\pm0.4}$ | -5.2$_{\pm0.4}$ | -4.4$_{\pm0.5}$ | **2.4**$_{\pm0.2}$ | **1.9**$_{\pm0.4}$ | -4.6$_{\pm0.4}$ |
| Complex | 0.7$_{\pm0.4}$ | **0.4**$_{\pm0.3}$ | 4.3$_{\pm0.2}$ | 1.3$_{\pm0.2}$ | 2.9$_{\pm0.2}$ | -0.5$_{\pm0.2}$ | -1.2$_{\pm0.0}$ | -1.9$_{\pm0.5}$ |
| Simple | -5.0$_{\pm0.5}$ | -9.0$_{\pm0.5}$ | -4.4$_{\pm0.5}$ | -3.9$_{\pm0.2}$ | -4.8$_{\pm0.4}$ | -3.9$_{\pm0.5}$ | -3.5$_{\pm0.4}$ | -1.8$_{\pm0.2}$ |
| V1 stoch. | **2.5**$_{\pm0.4}$ | 0.3$_{\pm0.3}$ | **12.0**$_{\pm0.5}$ | **2.6**$_{\pm0.3}$ | **10.0**$_{\pm0.8}$ | -2.8$_{\pm0.3}$ | -4.0$_{\pm0.7}$ | -4.0$_{\pm0.4}$ |

Table D.2: **Difference between the white box attack accuracies of the several VOneResNet50 variants and the unmodified VOneResNet50**. Removal of V1 stochasticity considerably reduced accuracy to all white box attacks. Removal of V1 distributions improved accuracy for the $L_\infty$ white box attack with high perturbation strength.

| | PGD-L1 | | PGD-L2 | | PGD-L$\infty$ | |
|---|---|---|---|---|---|---|
| Component removed | 40 [$\Delta\%$] | 160 [$\Delta\%$] | 0.15 [$\Delta\%$] | 0.60 [$\Delta\%$] | 1/1020 [$\Delta\%$] | 1/255 [$\Delta\%$] |
| V1 dist. | $-0.5_{\pm0.8}$ | $0.1_{\pm1.1}$ | $-0.8_{\pm0.2}$ | $0.6_{\pm1.0}$ | $0.3_{\pm0.3}$ | $\mathbf{3.0}_{\pm1.8}$ |
| Low SF | $-0.8_{\pm1.2}$ | $-3.1_{\pm1.1}$ | $-0.6_{\pm1.0}$ | $-2.3_{\pm0.8}$ | $-0.8_{\pm1.3}$ | $-1.8_{\pm1.0}$ |
| High SF | $-2.2_{\pm0.9}$ | $-4.7_{\pm2.6}$ | $-2.6_{\pm1.5}$ | $-5.1_{\pm1.2}$ | $-2.6_{\pm1.4}$ | $-5.5_{\pm1.1}$ |
| Complex | $-1.3_{\pm0.9}$ | $-7.4_{\pm1.6}$ | $-1.1_{\pm0.9}$ | $-7.2_{\pm1.2}$ | $-1.9_{\pm1.0}$ | $-7.9_{\pm1.2}$ |
| Simple | $-2.5_{\pm0.8}$ | $-1.4_{\pm1.4}$ | $-2.9_{\pm1.1}$ | $-1.7_{\pm1.2}$ | $-2.7_{\pm1.2}$ | $-0.6_{\pm0.8}$ |
| Noise | $-23.7_{\pm0.9}$ | $-36.5_{\pm0.8}$ | $-21.6_{\pm0.5}$ | $-40.1_{\pm0.4}$ | $-32.2_{\pm0.4}$ | $-29.2_{\pm0.4}$ |