[Reviews · NeurIPS 2020]

Review 1

Summary and Contributions: [UPDATE AFTER REBUTTAL] The authors have clarified a number of questions and provided some additional controls that aim at improving the probability of success of their adversarial attacks. While I'm still somewhat skeptical of the result and the lack of a clear explanation for *why* the VOneBlock would improve adversarial robustness, I'm giving them the benefit of the doubt. Since the authors state that they will release code, people will be able to verify the claims of the paper. I therefore support acceptance of the paper. [ORIGINAL REVIEW] The authors claim that prepending a fairly simplistic model of primary visual cortex to standard CNNs improves their robustness, measured as robustness against (1) adversarial attacks and (2) image corruptions. While corruption robustness increases only marginally, the effect is mainly driven by adversarial robustness. Here, the main driver of robustness appears to be stochasticity. These findings would be of substantial significance if they turned out to be true. However, given the large number of papers that have claimed to introduce methods for adversarial robustness in the past (including stochasticity) and have almost all been proven to be ineffective, I am very skeptical. Extraordinary claims require extraordinary evidence, but such evidence is unfortunately largely lacking.

Strengths: + The paper is well motivated, clearly written and easy to follow + The motivating result (Fig. 1) that adversarially more robust models are better models of V1 is interesting in its own right

Weaknesses: - Only one adversarial attach (PGD) is used, and not a particularly strong one - A number of controls to verify the efficacy of the attack are missing - No intuitive or mechanistic explanation *why* the result should be true is provided - Results are generally a bit overstated since improvements on corruption robustness are marginal and clean performance drops substantially

Correctness: The analyses are carried out fairly carefully, but a number of important controls are missing. Demonstrating adversarial robustness is notoriously difficult, since it cannot be proven and failure to find adversarial examples with any given method does not imply robustness. Since there are literally dozens of papers that have claimed robustness and have later been proven wrong, the bar for such claims should be very high and the burden of proof is on the authors to demonstrate they have taken all steps to maximize the probability of successfully finding adversarial examples. Here are a couple of suggestions what is missing from the current manuscript: - The authors use fairly small step sizes (eps/32), which tend to get trapped in local minima. There is no indication of whether they use random restarts or any other steps they may have taken to ensure they're not finding suboptimal solutions. - The results in Table B.1 are somewhat worrying: averaging gradients usually improves the success of attacks substantially. Given the low average activity (rate of 0.3) and the Poisson-like noise, the signal-to-noise ratio is very low. Thus, it is conceivable that the 10 samples the authors use are too few. They should also try larger numbers and verify that performance does not further improve by increasing the number of samples. - They use only PGD and no other attacks. If they stabilise the gradient sufficiently (see previous point), then they could easily use other white-box and black-box methods including DDN or Brendel & Bethge's attack. - For the control where they add only stochasticity to ResNet50 (Fig. 3B), I couldn't find any information on whether the authors scale the ResNet activations to be on the same scale (rate 0.3) as the VOneNet output and, if so, how. If this wasn't done, one possibility for the results in Fig. 3B would be that due to higher SNR the attack is more effective in ResNet+stochasticity than in VOneResNet.

Clarity: The paper is well written and easy to follow. For my taste, the authors refer to the Appendix too often and could try to include more data into the main paper to make it more self-contained by making the writing more concise at some places. For instance, the full page of discussion is rather uncommon and could be shortened substantially

Relation to Prior Work: Prior work is discussed appropriately. The authors did a very good job putting their work in context of a large body of previous work, in particular in neuroscience. With respect to adversarial robustness, the authors appear to be superficially aware of the issues surrounding evaluation and cite many of the relevant papers, but unfortunately don't act accordingly.

Reproducibility: Yes

Additional Feedback: Lines 51+52: The statement that "VOneBlock outperforms all standard ImageNet trained CNNs we tested at explaining V1 responses to naturalistic textures and noise samples" does not seem to be backed up by any data presented in the paper and sounds a bit odd given that it is essentially a Gabor filter bank as in Cadena et al 2019. I wonder if the authors could elaborate on this statement. Line 166: Please be specific about what you did rather than just referring to "key controls" and supplementary figures. Table 1: What do the +/- refer to? SEM, SD, CI? The paper does not include a statement about availability of code, which is unfortunate.


Review 2

Summary and Contributions: Develop a "hybrid" neural net with a fixed weight front-end that is meant to reflect primate V1 (early visual cortex) as input to standard CNNs. The hybrid network is more robust to image perturbations.

Strengths: Interesting paper that follows in a spate of recent work in which "fixed" more traditional visual processing routines are used as precursors to CNNs. Sound idea, that is becoming popular. This is a well-done version in that it grounds very carefully in V1 models. And demonstrates a benefit.

Weaknesses: Not as novel as characterized. Also, it is not clear that human robustness to image perturbations arises from V1 processing. There is some effort to understand the source of the robustness, but better baselines may be warranted, that is, considering more specifically possible sources based on the biology. More generally, this seems to be one of many "lets put vision back in vision models" papers. This is an important point, but should be acknowledged in the context that CNNs are NOT vision SYSTEMS, but task-specific networks. Human vision is comprised on many different computational components, only a small subset of which are ever captured by a CNN - so really, a more robust, "human-like" vision system would need to include many of these components. As such, it would be useful to discuss such efforts and the more general issue of how that sort of architecture might be realized.

Correctness: Claims and methods are reasonable. Good overall, empirical methodology, but more baselines should be used to compare the V1 fixed weight front-end.

Clarity: Well written.

Relation to Prior Work: More on other models in which fixed-weight networks or straight-up image processing, etc. are used as front-ends should be cited/discussed (e.g., Simoncelli)

Reproducibility: Yes

Additional Feedback:


Review 3

Summary and Contributions: [UPDATE AFTER REBUTTAL] The authors addressed most of my concerns in their rebuttal. After reviewing the rebuttal and the other reviews, I am increasing my score. [ORIGINAL REVIEW] The authors begin by making a striking observation: that CNNs trained on ImageNet that better explain primate V1 recordings are also more robust to white box adversarial attacks. From this observation, they propose the following hypothesis: that a linear-nonlinear plus noise model of V1, when used in place of the first layer in CNN architectures, is sufficient to confer robustness to adversarial attacks. Put another way, this paper suggests that the reason the more V1-like CNN models are robust is due to something about the linear, nonlinear, and noise properties of the first layer. The authors test this hypothesis by replacing the first layer in a few different popular CNN architectures with a V1 like linear+nonlinear+noise layer, and shows that this improves adversarial robustness. Finally, the authors ablate different components of this layer (the linear, nonlinear, and stochastic pieces) to see which of these has the greatest effect on a merged benchmark of common corruptions and adversarial perturbations.

Strengths: I found the initial observation (that V1 explanatory power correlates with adversarial robustness) quite striking. Has this been shown before? What about correlation with other layers in the visual hierarchy? Is this true for mouse V1, in addition to primate V1? Is the CNN layer that is most correlated with V1 always the first layer, for each of these models? The authors perform the important experiment of ablating various components of the V1-block, to see how removing those pieces changes the robustness results. Overall, the paper is clear and well written.

Weaknesses: My main concern is that I think the paper could go further in understanding what particular mechanisms underlie the improved robustness. Let me explain: The first observation (Fig 1) demonstrates that CNNs that are better at describing V1 responses are more robust to adversarial attack. Then, the paper proposes that this is due to particular mechanisms in V1: linear filtering by a Gabor filter bank, followed by a nonlinearity, and noise. If this is true, then adding these mechanisms in CNNs should both: (1) improve their ability to be robust to adversarial noise, *and* (2) be better at describing V1 responses. The paper sort of tests point (1), although using a benchmark that combines adversarial perturbation with common corruptions. I think these two tests should be separated. Fig 1 should show the correlation between models that explain V1 with: adversarial robustness *and* robustness to common corruptions. If both are strongly correlated, that would greatly strengthen the arguments for the combined benchmark. If *only* adversarial robustness is correlated with V1 explainability, then having the combined benchmark obscures the real phenomenon we are trying to understand. Regarding point (2), the paper should show whether adding the V1 block improves the ability of the CNNs to describe V1 responses. Really, the V1 block alone should do a better job of describing V1 responses than any layer in any CNN model. If not, then that suggests that the added "V1 block" is not really like V1, but instead just a useful model front-end for improving robustness. The fact that this experiment is missing is a glaring omission. This is one benefit of the approach of Li et al, which _directly_ encourages the CNN to mimic neural representations. Related to this, I think the paper should score the ablation test (where individual pieces of the LNP are removed) on adversarial robustness and common corruptions _separately_. My suspicion is that the noise is what helps the models be robust to adversarial perturbations, and the Gabor filter bank is what helps the models be robust to common corruptions (this is observed in Table 2, but Figure 3 obscures this by only showing the combined average score). The reason why I think this is significant, is that it is unclear to me whether these properties are really unique to V1. As the paper mentions, other studies have observed that adding noise can improve adversarial robustness. And it seems equally plausible to me that having the large receptive fields in the Gabor filter bank would help average over specific types of noise in the common corruptions dataset. This, for me, gets at the real mechanisms underlying the phenomenon: additive noise and large receptive fields. (I know the paper tested robustness after adding noise directly in a CNN, but it did so _after_ the first layer, I would also be interested in seeing an experiment with injected noise directly in the inputs). All of this is to say: I think the paper could provide more experiments (outlined above) to alleviate concerns that the observed robustness improvements are really due to something unique about V1.

Correctness: The authors sample Gabor filters by taking measurements from the literature from a particular eccentricity, of 8 degrees? If I understand correctly, this would correspond to presenting images to the CNN at a particular resolution? Is this what is done?

Clarity: Yes, the paper is well written.

Relation to Prior Work: It seems some discussion of whether humans really are robust to white box adversarial attacks is warranted (http://papers.nips.cc/paper/7647-adversarial-examples-that-fool-both-computer-vision-and-time-limited-humans). In addition, given that LNP models explain a tiny fraction of variance in V1 responses to natural scenes (c.f. https://www.jneurosci.org/content/25/46/10577 or http://www.rctn.org/bruno/papers/V1-chapter.pdf), the paper could elaborate on these limitations in their discussion.

Reproducibility: Yes

Additional Feedback:


Review 4

Summary and Contributions: This paper presents a biologically motivated defense against adversarial attacks. The paper first observes that the robustness of convolutional neural networks is correlated with the ability of their activations to explain the recorded responses of primate V1 neurons. The main contribution of the paper however is a new "prefix" architecture that reduces the effectiveness of adversarial attacks while minimally degrading accuracy on unperturbed images. This new architectural block is drawn from existing models of V1 neurons, and is systematically ablated to identify what components are critical for the adversarial defense.

Strengths: This paper is impactful, thorough, and well written. Despite active research in the field of adversarial attacks, state-of-the-art defenses involve computationally intensive adversarial training. In contrast, this paper reports a relatively simple module - with fixed weights - that dramatically reduces the effectiveness of an adversarial attack. The analysis of how this result arises is carefully examined and analyzed across many types of different convolutional neural networks (AlexNet, VGG, ResNet, ResNeXt, DenseNet, SqueezeNet, ShuffleNet, MnasNet). Unusually for the field, standard deviation error bars are shown. The paper takes care to ensure that the vector of attack has proper white-box access to the new proposed module, using the reparameterization trick to attack the stochastic elements. This addressed my primary concern that the new module was simply concealed from the attack.

Weaknesses: The main weakness in my opinion is the effectiveness of the adversarial attack method the authors chose. In particular, the defense against adversarial attacks is largely due to the stochastic element (Fig. 3A). I was uncertain how the Monte Carlo estimates would affect the accuracy of the projected gradient descent steps. The effectiveness of the adversarial attack on this new module hinge on the accuracy of these gradient approximation methods, and it is difficult to ascertain from the paper what the quality of this approximation is and how the validity of the attack varies with the approximation quality.

Correctness: The methods used to determine neural explained variance seemed correct and appropriately split the neural data into train and test phases. The authors additionally appeared familiar with the adversarial attack literature and aside from my comments above I did not see a flaw in their approach or method.

Clarity: The paper is well written and clear. At times the paper glosses over underlying complexity in the method - for instance only a few sentences are allocated to the V1 neural predictivity methods - but the questions that arise are largely addressed in supplemental materials or in cited work.

Relation to Prior Work: Prior work is described concisely and adequately contrasted with the paper's contributions.

Reproducibility: Yes

Additional Feedback:

[Author Response · NeurIPS 2020]

We thank the reviewers for their constructive comments. We are pleased that even the most critical reviewers found
the claims to be of "substantial significance" *if* they are validated with key controls. *We have completed all of those*
*controls*; while some were in the original submission but not sufficiently highlighted, others are new analyses following
reviewer suggestions. In all cases, the controls support our claims and we would happily include these improvements in
the final manuscript. We thank the reviewers for their willingness to reconsider their reviewer score in this light.

**VOneBlock describes V1 responses better than CNNs.** We agree with R1 and R3 that
*if* the VOneBlock is not better than CNNs at describing V1 responses, it would undermine
our claims. In lines 155-158 we write "the VOneBlock outperformed all tested ImageNet-
trained CNNs in explaining responses in the V1 dataset used", and provided the value
of explained variance (0.387±0.007). Here, we update Fig.1, showing VOneResNet50
surpasses all standard ImageNet-trained CNNs. We are confident in this result since we
searched over all layers in a large pool of CNNs and did not find any coming close to
the V1 explained variance of the VOneBlock. Further, **our results are consistent with**
**Cadena et al 2019**: our GFB has parameters constrained by empirical data resulting in
a better model of V1; when we use the parameters of the GFB in Cadena et al, we obtain
a much lower explained variance (0.296±0.005, marked on x-axis of Fig.R1).

Fig R1: Accuracy in weak white box attack vs V1 explained variance; VOneBlock outperforms standard CNNs

**Stochasticity during the attack is not the main source of adversarial robustness.** R1 and R4 are concerned that
stochasticity makes VOneNets artificially appear more robust. However, the majority of the VOneNets' robustness does
not originate from stochasticity during the attacks, but rather from training downstream layers with V1-like features and
neural stochasticity. *When we turn off stochasticity completely during attack/inference, VOneResNet50 retains* 70% *of*
*the adversarial robustness gains of the model with stochasticity on, and significantly outperforms ResNet50 (Fig.4).*
Meanwhile, when we add stochasticity to ResNet50, using an affine transformation to scale the activations to the same
magnitude as in the VOneBlock (lines 148-151), we find substantially smaller improvements in robustness (Fig.3B).

**Further adversarial attack optimization does not overturn results.** We follow
R1's suggestions to further verify our attacks, addressing R4's concerns of gradient
quality as well. Increasing the attack step size ($\alpha$) and the number of gradient
samples ($k$) only marginally increases the attack effectiveness. A grid search over
$\alpha$ and $k$ (Fig.R.2; PGD with $\|\delta\|_\infty = 1/255$) reveals that at the most effective step
size ($\alpha = \epsilon/8$), increasing gradient samples beyond $k = 16$ no longer improves
the attack. At $k = 128$, VOneResnet50 accuracy is only reduced from 29.1% to
26.0%, remaining a large margin above ResNet50 at 0.8%. We do not find using
5-10 random restarts improves beyond the most effective $k$ and $\alpha$ settings. Thus,
these controls do not qualitatively change our results. Finally, because our model
has a stochastic boundary, we focus on PGD and avoid boundary targeted attacks,
as the most effect should come from staying as far from the boundary as possible.

Fig R2: Attack optimization only marginally decreases VOneResNet50 accuracy

**Common corruptions provide additional insights.** R1 and R3 question the in-
clusion of common corruptions. While common corruption scores generally do not
correlate with V1 similarity, we include them because robustness to imperceptible adversarial attacks should not come
at the cost of performance on other perturbations that humans easily deal with, as is the case for adversarially trained
models which underperform on this benchmark relative to the baseline and VOneNets. Further, common corruptions
provide insights into the role of different components in the VOneBlock in dealing with specific stationary perturbation
statistics. **Common corruptions and white box attacks are properly separated**: results are shown separately in
Tables 1, 2, B.2, C.4, and C.5; white box only in Fig.1, 2, and 4; and scores are only combined in Fig.3.

**Mechanisms contributing to robustness.** We agree with R1 and R3 that the mechanisms behind our results warrant
further exploration, but believe our study provides a novel and important characterization of how low level visual
processing relates to robust object recognition behavior. With ablations, we characterize how different components of
the VOneBlock impact performance on a variety of perturbation statistics. Lines 212-218 discuss the effect of high
SF Gabors on various corruptions as well as complex cells on adversarial attacks, and Fig.4 shows that the improved
adversarial robustness largely stems from training downstream layers with a stochastic V1 front-end.

**Other questions from R1 and R3.** ± refers to SD. As stated in the appendix, we will release code and model weights.
To the best of our knowledge, the correlation between adversarial robustness and V1 similarity has not been shown
before. The CNN layer that best predicts V1 responses is usually not the first (except for the VOneNets). We used
parameters from the foveal region of V1 (eccentricity < 5deg). Our model has a field-of-view of 8deg (resolution =
224px/8deg; eccentricity < 4deg). The VOneBlock and all tested CNNs explained less than 40% of V1 response variance
which is in line with the literature and is a possible leeway to make progress on robustness. One of the comparison
models (ANT3×3+SIN) is trained with input noise and does not show improvements in adversarial robustness.

[Meta-Review · NeurIPS 2020]

In this submission, the authors present a hypothesis that better aligning models with computations within the primary visual cortex leads to more robust models. The authors first show that model robustness is correlated with the ability of a CNN to explain the variance in primary visual cortex recordings. Based on this observation, the authors develop a hybrid model for image classification in which a CNN is prepended with fixed features from a simple model of primary visual cortex. The authors demonstrate that the resulting hybrid model is more robust to white box adversarial attacks and marginally more robust to image distortions. Through a series of ablations the authors attribute the gains in robustness to the nonlinearity and stochasticity -- although the stochasticity provides the greatest benefit. The reviewers offer some concerns about the strength of the overall methods for assessing robustness. That said, all authors found the presentation of the results extremely clear and the line of research quite exciting. For all of these reasons, this paper will be accepted and published at NeurIPS.